# Orally Administered 6:2 Chlorinated Polyfluorinated Ether Sulfonate (F-53B) Causes Thyroid Dysfunction in Rats

**DOI:** 10.3390/toxics8030054

**Published:** 2020-08-08

**Authors:** So-Hye Hong, Seung Hee Lee, Jun-Young Yang, Jin Hee Lee, Ki Kyung Jung, Ji Hyun Seok, Sung-Hee Kim, Ki Taek Nam, Jayoung Jeong, Jong Kwon Lee, Jae-Ho Oh

**Affiliations:** 1Toxicological Research Division, National Institute of Food and Drug Safety Evaluation, Ministry of Food and Drug Safety, Osong, Chungcheongbuk-do 28159, Korea; shhong99@korea.kr (S.-H.H.); seunghee88@korea.kr (S.H.L.); yangjy@korea.kr (J.-Y.Y.); tod98@korea.kr (J.H.L.); kikyung@korea.kr (K.K.J.); seokjh@korea.kr (J.H.S.); 0jjy@korea.kr (J.J.); jkleest@korea.kr (J.K.L.); 2Severance Biomedical Science Institute, College of Medicine, Yonsei University, Seoul 03760, Korea; ksh0814@yuhs.ac (S.-H.K.); kitaek@yuhs.ac (K.T.N.)

**Keywords:** F-53B, subchronic oral toxicity, thyroid hormone, thyroid dysfunction

## Abstract

The compound 6:2 chlorinated polyfluorinated ether sulfonate (F-53B), a replacement for perfluorooctanesulfonate (PFOS) in the electroplating industry, has been widely detected in numerous environmental matrices, human sera, and organisms. Due to regulations that limit PFOS use, F-53B use is expected to increase. Therefore, in this study, we performed a subchronic oral toxicity study of F-53B in Sprague Dawley (SD) rats. F-53B was administered orally once daily to male and female rats for 28 days at doses of 5, 20, and 100 mg/kg/day. There were no toxicologically significant changes in F-53B-treated rats, except in the thyroid gland. However, F-53B slightly reduced the serum concentrations of thyroid hormones, including triiodothyronine and thyroxine, compared with their concentrations in the vehicle group. F-53B also induced follicular hyperplasia and was associated with increased thyroid hormone biosynthesis-associated protein expression. These results demonstrate that F-53B is a strong regulator of thyroid hormones in SD rats as it disrupts thyroid function. Thus, caution should be exercised in the industrial application of F-53B as an alternative for PFOS.

## 1. Introduction

Perfluoroalkyl and polyfluoroalkyl substances (PFASs) are a class of synthetic organic compounds with broad applications in industrial substances and consumer products [1]. PFASs are bioaccumulative and persistent in the environment, and their carbon–fluorine bond confers heat and chemical stability. These properties, particularly of long-chain substances (≥C6), observed in laboratory animal studies are of considerable concern [2].

In particular, perfluorooctanesulfonate (PFOS), perfluorooctanoic acid (PFOA), and their salts have attracted worldwide attention in the regulatory community [3]. PFOS has several toxic effects in rodent studies, including hepatotoxicity, neurotoxicity, and reproductive toxicity, and disrupts thyroid function [4,5,6,7]. Due to its toxicity and potential carcinogenicity, PFOS was added to the Stockholm Convention on Persistent Organic Pollutants in 2009 [8]. Subsequently, studies have provided epidemiological evidence of the human health risks of PFOS [7,9]. However, due to the lack of a suitable replacement, one of the exceptions in the convention for the production and use of PFOS was for the metal-plating industry. Therefore, PFOS has continued to be used in the electroplating industry, for food contact materials and to coat frypans [10].

Due to its similar structure to PFOS, 6:2 chlorinated polyfluorinated ether sulfonate (6:2 Cl-PFAES; Cl-C_6_F_12_-O-CF_2_CF_2_SO_3_) has been used as an alternative (Figure 1). Potassium 6:2 Cl-PFAES is the major component of the commercial goods F-53B (trade name). In China, F-53B is accepted for use in the electroplating industry. A paper by the China Metal-Plating Association revealed an annual usage of 30–40 tons [11,12]. F-53B has been found in Arctic wildlife (0.023–0.27 ng/g) and the rivers of Western countries (0.01–0.38 ng/L), indicating that it may have become a global pollutant [13]. In addition, recent research demonstrated the high prevalence of F-53B components in human biomonitoring samples in China, and 6:2 Cl-PFESA has been detected in river water, human serum, and fish muscle [14,15]. Furthermore, in an analysis of human serum and urine samples, Cl-PFESA was the most persistent PFAS; therefore, the toxic effects of 6:2 Cl-PFESA are expected to be large [16].

Recently, the National Toxicology Program presented toxicity reports of long-chain PFASs [17,18]. The reports demonstrated that PFASs with PFOA and perfluoroalkyl carboxylates induced changes in the serum thyroid hormone levels and relative thyroid weight. In addition, high-dose PFOA-treated rats exhibited thyroid gland follicular hypertrophy [17]. PFASs with PFOS and perfluoroalkyl sulfonates also induced changes in serum thyroid hormone concentration [18]. F-53B, however, was not included in these reports. In previous studies, F-53B altered thyroid hormones in adult zebra fish and offspring larvae [19]. F-53B has greater agonistic activity on thyroid hormone transport proteins than PFOS in rat pituitary cells [20]. However, these results suggest only indirect evidence on thyroid hormone regulation and there is not a toxicological database on thyroid dysfunction in rodents.

Despite its potential thyroid toxicity, studies on the toxic effects of F-53B are relatively limited. Therefore, the aims of this study were to analyze the effects of F-53B on thyroid functions and to initiate an in vivo toxicological database for F-53B.

## 2. Materials and Methods

### 2.1. Test Substance

F-53B (CAS No.: 73606-19-6, purity ≥ 99%) was obtained from ALFA Chemistry Protheragen Inc. (Ronkonkoma, NY, USA). The compound was dissolved in dimethyl sulfoxide (DMSO, purity > 99.9%, Sigma-Aldrich, St. Louis, MO, USA), to prepare a stock solution, and the working solution was obtained by dilution in distilled water. The final concentration of DMSO was 0.01% in both vehicle control and treatment groups [21]. Antibodies for thyroid-stimulating hormone receptor (TSHR; sc-53542) and thyroperoxidase (TPO; sc-58432) were obtained from Santa Cruz Biotechnology (Dallas, TX, USA).

### 2.2. Experimental Design

Eight-week-old male and female specific pathogen-free Sprague Dawley rats (Crl:CD (SD); *n* = 7 per sex and group) were purchased from Koatech (Seoul, Korea) for use in the general toxicity studies. The rats were acclimated for 7 days after arrival at the laboratory of the National Institute of Food and Drug Safety Evaluation of the Ministry of Food and Drug Safety (MFDS; Osong, Korea), then randomly divided into groups. The animals were housed at a temperature of 22 °C ± 3 °C and a relative humidity of 50% ± 20%. The animal facility was maintained under a 12-h light/dark cycle (from 8.00 a.m. to 8.00 p.m.) with 10–20 air ventilations per hour. The animals were allowed access to feed (LabDiet 5002; PMI Nutrition, Richmond, VA, USA) and water (autoclaved water) ad libitum. The 28-day repeated-dose oral toxicity study was carried out according to the “Toxicity Test Standards for Drugs” of the Korean MFDS. F-53B (5, 20, and 100 mg/kg/day) was orally administered once daily for 28 days. The administration (from 9.30 a.m. to 11.00 a.m.) volume for all animals was 10 mL/kg based on the most recent individual body weight. Body weights were measured twice a week during the treatment period, on the day of grouping, and upon necropsy. Necropsy and the administration of male rats were performed before one day. Experimental protocols involving animals in this study were reviewed by the Institutional Animal Care and Use Committee of the MFDS (identification code: MFDS-19-015; date of approval: 23 January 2019).

### 2.3. Clinical Pathology and Hormone Measurement

Approximately 0.5 mL of whole blood was collected for hematology and 1.5 mL of blood was collected into serum separator tubes for clinical chemistry and the analysis of serum thyroid hormone levels. Blood samples were collected in a round robin (zigzag) manner from 9.00 a.m. to 12.30 a.m. The blood was placed in 3-mL Vacutainer^®^ tubes (BD, Franklin Lakes, NJ, USA) containing the anticoagulant EDTA-2K. The blood samples were coagulated by incubation at room temperature for 30 min, and were then centrifuged for 10 min (3000 rpm) to obtain serum. The hematology parameters—red blood cell, white blood cell, platelet, neutrophil, lymphocyte, monocyte, eosinophil, basophil, large unstained cell, and reticulocyte counts; hemoglobin and hematocrit levels; and mean corpuscular volume, mean corpuscular hemoglobin, and mean corpuscular hemoglobin concentration—were measured with an automated cell counter (ADVIA 2120; Siemens, Munich, Germany). Using a TBA-200FR NEO chemistry analyzer (Toshiba, Tokyo, Japan), we assessed the following clinical chemistry parameters: alanine aminotransferase, aspartate aminotransferase, alkaline phosphatase, gamma glutamyl transferase, blood urea nitrogen, creatinine, total protein, total bilirubin, total cholesterol (TCHO), glucose, triglyceride, albumin, albumin/globulin ratio, calcium, sodium, chloride, potassium, phosphorus, high-density lipoprotein (HDL), and low-density lipoprotein (LDL) levels. Triiodothyronine (T3), thyroxine (T4), and thyroid-stimulating hormone (TSH) were measured with a IMMULITE^®^ 1000 analyzer (Siemens).

### 2.4. Necropsy and Measurement of Organ Weight

After necropsy, we collected the following tissues for gross observation and histological analysis: liver, kidneys, adrenal glands, brain, heart, thyroid gland, pituitary gland, spleen, thymus, testes, epididymis, prostate, seminal vesicle, coagulating gland, ovaries, and uterus. Necropsy body weights along with absolute organ weights were obtained, and organ-to-body weight ratios were calculated. Representative samples of the collected organs were fixed in 10% neutral buffered formalin, except for the testes and epididymis, which were fixed in modified Bouin solution.

### 2.5. Histological Analysis and Protein Expression

All tissues were fixed in 10% formalin overnight or modified Bouin solution, as described above, then embedded in paraffin wax, routinely processed, and sectioned into 4-μm-thick slices. The tissue sections were de-paraffinized and then rehydrated in a descending graded series (100%, 95%, and 70%) of ethanol. We performed antigen retrieval (Dako S1699, Agilent Technologies, Santa Clara, CA, USA) using a pressure cooker. The sections were incubated in 3% H_2_O_2_ in PBS for 30 min to block endogenous peroxidase activity. The sections were then washed twice with phosphate-buffered saline (PBS) and incubated with Protein Block, Serum-Free (Dako, X0909) for 1–2 h at room temperature to minimize non-specific antibody binding. We treated the sections with M.O.M.^®^ reagent (BMK-2202, Vector Laboratories, Burlingame, CA, USA) for 1 h when using mouse primary antibodies. The sections were incubated with primary antibodies overnight at 4 °C. After three washes in PBS, the sections were incubated with horseradish peroxidase-conjugated secondary antibodies (Dako K4003 and K4001, Agilent Technologies) for 15 min at room temperature. For immunohistochemistry, we developed the antibody signal with DAB (3, 3-diaminobenzidine) (Dako K3468, Agilent Technologies) and counter-stained with Mayer’s hematoxylin (Dako S3309, Agilent Technologies). Five rats were analyzed in each group and DAB-positive cells were quantified in each slide. DAB-positive cells were counted in entire thyroid and one slide per one rat was analyzed. The scanned image was analyzed by the number of DAB-positive cells per total area using the digital pathology software, QuPath (https://qupath.github.io/).

### 2.6. Statistical Analysis

The results are presented as mean ± standard deviation (s.d.). All data were analyzed using a one-way analysis of variance followed by a post hoc Dunnett’s multiple comparison test. The statistical analyses were performed using GraphPad Prism. Differences with a *p*-value < 0.05 were considered statistically significant.

## 3. Results

### 3.1. Thyroid Hormones and Thyroid Hormone-Related Parameters

After treating the rats with 0, 5, 20, or 100 mg/kg/day of F-53B for 28 days, we measured their serum thyroid hormone levels. Total T3 and T4 serum concentrations were significantly lower in all treated groups, in a dose-dependent manner, than in the vehicle control group. The total T3 and T4 levels were two-fold lower in response to F-53B treatment (Figure 2A,B). However, the TSH concentration was not changed by F-53B administration compared with vehicle treatment in male or female rats (Figure 2C). The thyroid hormone-associated parameters, including TCHO, HDL, and LDL levels, did not differ among the control and treated groups (Figure 2D–F).

### 3.2. Body and Organ Weights

There was a moderate increase in body weight in the F-53B treatment groups at day 28 that did not reach statistical significance (Figure 3A,B). There were no significant differences among the relative thyroid weights of the male or female rats in all groups (Figure 3C,D).

### 3.3. Histopathology

After treatment with F-53B, the rats exhibited largely normal thyroid histological features after exposure to F-53B. However, we found treatment-related follicular hyperplasia in the thyroid glands at doses of F-53B ≥ 5 mg/kg/day (Table 1; Figure 4). In particular, we observed a higher incidence of more severe follicular cell hyperplasia in the female high-dose F-53B group. There were no treatment-related histopathological changes in the other tissues in any of the groups.

### 3.4. Thyroid Hormone Biosynthesis-Related Protein Expression

To evaluate the mechanism of thyroid hormone regulation, we investigated the expression of thyroid hormone biosynthesis-related proteins in the rat thyroid gland by immunohistochemistry. TSHR expression was significantly higher in the high-dose (100 mg/kg/day) F-53B group than in the vehicle group. The protein expression of TPO also tended to be higher after F-53B treatment (Figure 5).

## 4. Discussion

Thyroid hormones play critical roles in differentiation, growth, and metabolism. Thyroid hormone secretion and synthesis are regulated by a negative feedback system that includes the hypothalamic–pituitary–thyroid (HPT) axis. TSH secreted by the pituitary gland is the primary regulator of thyroid hormone release and secretion [22]. Several genes play roles in the biosynthesis of thyroid hormones, such as thyroid stimulating hormone receptor (TSHR), thyroid peroxidase (TPO), thyroglobulin (Tg), and sodium iodide symporter (SLC5A5), in the thyroid follicle [23,24,25].

In 2018, the Organization for Economic Co-operation and Development test guidelines (OECD TG 408) were revised to add thyroid hormone measurement to improve the detection of potential endocrine toxicity [26]. The revised guideline requires the measurement of T4, T3, and TSH serum levels, and the weight of the thyroid gland, all of which are responsive to thyroid pathway perturbation. Besides, serum TCHO, HDL, and LDL levels should be determined as the levels of these parameters are directly controlled by thyroid hormone action, which reports thyroid function.

In the present study, a toxicity study was performed after 28-day repeated oral administration of F-53B in SD rats. There were no toxicologically significant changes with regard to organ weight change or hematological and biochemical examination. The statistically significant changes were accompanied by parameter-associated changes in the histopathological examination or were not dose-dependent (Appendix A). However, we found that the concentrations of T3 and T4 were significantly lower in F-53B–treated rats in the absence of TSH changes. The reason for a lack of TSH response in the face of substantially lower thyroid hormone levels is not clear, nor is it consistent with a disruption of the HPT axis. However, the few known studies evaluating the effects of the oral administration of PFASs on thyroid hormone status also reported decreases in thyroid hormone levels without compensatory increases in TSH [27,28]. We confirmed the development of thyroid follicular hyperplasia via the histopathological examination of F-53B-treated rats, as well as increased protein expression of TSHR and TPO by immunohistochemistry. These results indicate a compensatory response to the marked decrease in serum thyroid hormones, such as T3 and T4, that is consistent with the findings of other studies [17,18].

In a previous study, PFOS reduced the circulating T3 and T4 levels, with no effect on serum TSH in rats [18]. Another study demonstrated that PFOS increased the proliferation of thyroid follicular cells compared the control group [29]. These findings indicate that PFOS not only has a similar structure but also induces a similar response to F-53B in an in vivo system. Similarly, PFOA induced a decrease in serum thyroid hormone concentration and relative thyroid weight, but PFOA induced hyperplasia of the thyroid gland in a histological examination [17]. Indeed, data regarding some PFASs are contradictory, and hypothyroxinemia studies have reported a sexually dimorphic response pattern to PFASs [27,28,29,30]. Recent research has demonstrated that parental exposure resulted in F-53B transfer as well as an increase in T4 concentration in zebrafish. The significant increase in T4 and decrease in T3 were accompanied by altered transcriptional levels of the HPT axis [19]. Those results differ from ours, as we observed lower thyroid hormone concentrations in F-53B-treated SD rats than in controls. This discrepancy may arise from species-specific mechanisms of the F-53B-mediated regulation of the HPT axis. F-53B may also function more sensitively in rat thyroid glands than in zebrafish.

## 5. Conclusions

We performed a 28-day subchronic oral toxicity study of F-53B (5, 20, and 100 mg/kg/day) in SD rats. The present study demonstrated that F-53B induces thyroid dysfunction in rats, characterized by reduced T3 and T4 levels, thyroid follicular hyperplasia, and abnormal thyroid hormone biosynthesis-related protein expression. Further studies on the consequences of these molecular mechanisms are required to fully determine the risks associated with the use of F-53B. Our results suggest that F-53B is a powerful regulator of thyroid hormones in rats as it disrupts thyroid function.

## Figures and Tables

**Figure 1 toxics-08-00054-f001:**
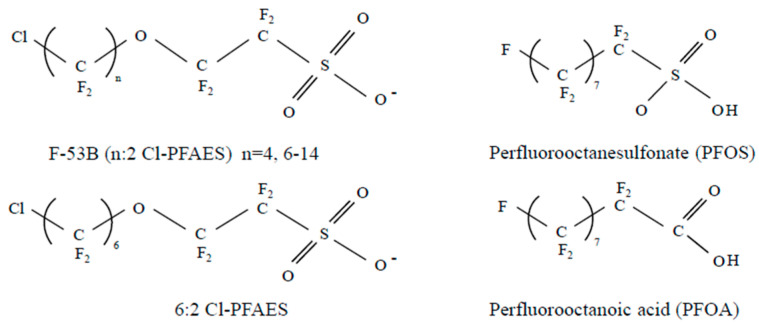
Molecular structural differences between C8-PFASs. All chemicals have eight carbons. Differences between 6:2 Cl-PFAES and PFOS are ether residue. PFOS and 6:2 Cl-PFAES contain in sulfonate and PFOA includes carboxylic acid residue.

**Figure 2 toxics-08-00054-f002:**
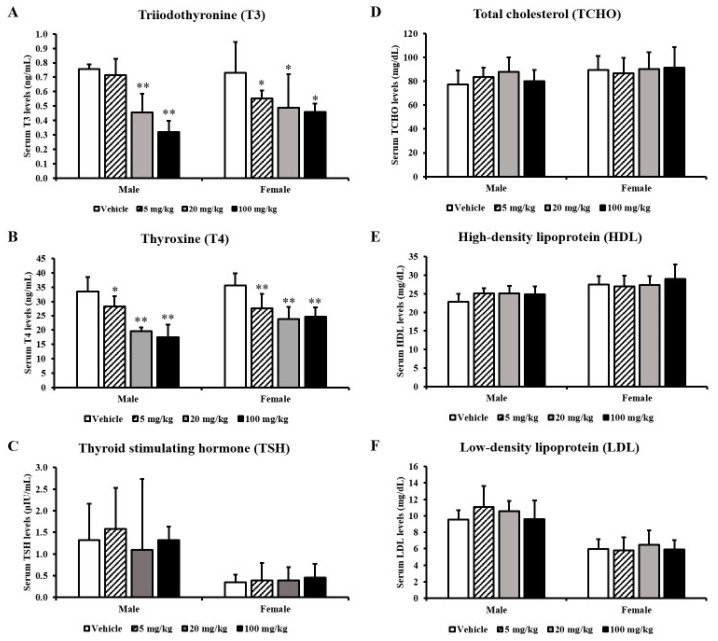
Serum concentrations of thyroid hormones and hormone-related parameters in rats treated with F-53B. The levels of triiodothyronine (T3) (**A**), thyroxine (T4) (**B**), thyroid-stimulating hormone (TSH) (**C**), total cholesterol (TCHO) (**D**), high-density lipoprotein (HDL) (**E**), and low-density lipoprotein (LDL) (**F**) in rat serum after F-53B treatment. The data are displayed as the mean ± s.d. (*n* = 7). For values that differ significantly as assessed by one-way analysis of variance and Dunnett’s test: * indicates *p* < 0.05 compared with the vehicle control group and ** indicates *p* < 0.01 compared with the vehicle control group.

**Figure 3 toxics-08-00054-f003:**
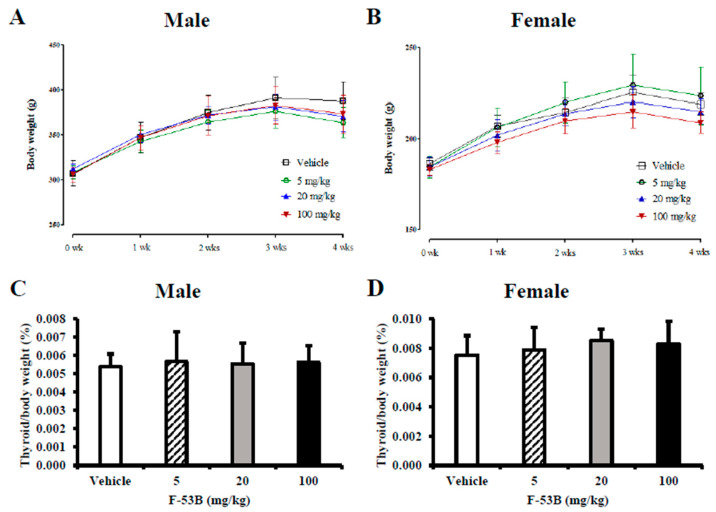
Body and relative organ weights after F-53B administration. Body weight changes after F-53B administration for 28 days in male (**A**) and female (**B**) rats. The thyroid/body weight ratios are shown for male (**C**) and female (**D**) rats. The data are displayed as the mean ± s.d. (*n* = 7). For values that differ significantly as assessed by one-way analysis of variance and Dunnett’s test: * indicates *p* < 0.05 compared with the vehicle control group and ** indicates *p* < 0.01 compared with the vehicle control group.

**Figure 4 toxics-08-00054-f004:**
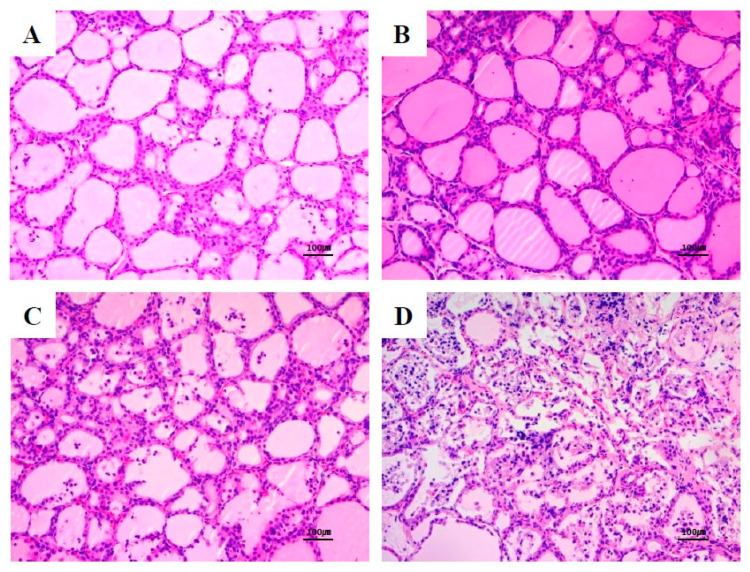
Follicular hyperplasia in the thyroid glands of female Sprague Dawley rats exposed to 0 (vehicle) (**A**), 5 mg/kg/day (**B**), 20 mg/kg/day (**C**), and 100 mg/kg/day (**D**) F-53B by oral gavage for 28 days.

**Figure 5 toxics-08-00054-f005:**
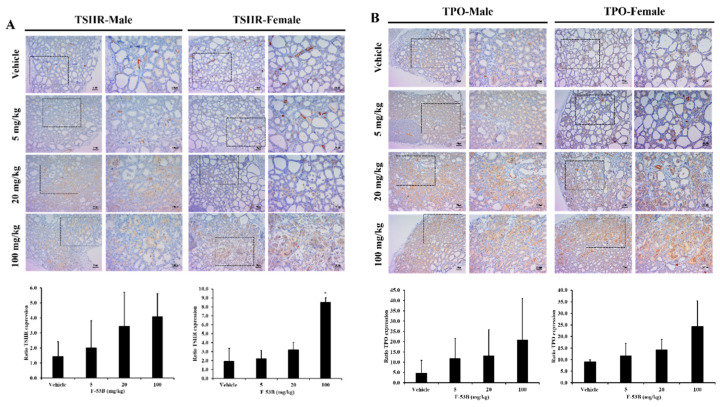
Immunostaining of thyroid hormone biosynthesis-associated proteins in the thyroid gland tissue of F-53B-treated rats. Thyroid tissues were stained with thyroid-stimulating hormone receptor (TSHR) (**A**) and thyroperoxidase (TPO) (**B**) specific antibodies to confirm the localization and expression of the proteins in male and female rats. The data are displayed as the mean ± s.d. For values that differ significantly as assessed by one-way analysis of variance and Dunnett’s test: * indicates *p* < 0.05 compared with the vehicle control group.

**Table 1 toxics-08-00054-t001:** Incidences of thyroid gland follicular hyperplasia in the 28-day toxicity study of F-53B.

Organs	Groups (mg/kg/day) ^a^			
	Vehicle	5	20	100
Male				
Thyroid gland				
Hyperplasia	0	1	3	1
Female				
Thyroid gland				
Hyperplasia	0	0	4	5

^a^ Thyroid gland tissues were examined five rats in dosing group respectively.

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
