# Peer review of "Orally Administered 6:2 Chlorinated Polyfluorinated Ether Sulfonate (F-53B) Causes Thyroid Dysfunction in Rats"

_toxics, 2020, doi:10.3390/toxics8030054_

Round 1

Reviewer 1 Report

This manuscript by Hong et al. is a straightforward 28-dosing study with the PFAS polyfluorinated ether sulfonate (F-53B). Specifically, the authors present data regarding how F-35B impacts serum thyroid hormones as well as thyroid gland histopathology. F-53B is not yet widely studied and thyroid is one of the most well-supported endpoints of PFAS exposure, so this work could have a positive impact in the field. However, some methods were missing, Figure 2 is not legible, and the discussion could use a more thorough synthesis of the literature.

Major Comments

Lines 35-36 – I would clarify what species the authors are referring to for these statements, and it would also be helpful to include any epidemiological data to support the animal studies.

Lines 49-50 – It is suggested that the statement regarding the bioaccumulation potential of 6:2 Cl-PFESA be directly discussed and defended. Is this hypothesized given the chemical’s human half-life? It’s environmental prevalence?

Lines 51-52 – Please provide a link or citation regarding the NTP toxicity reports.

Line 53 – I think the authors are referring to how PFOS and PFHxS are thyroid disruptors in the NTP report? But this link wasn’t totally clear. I would clarify this sentence and name specific PFASs.

Line 63 – Please state the purity and manufacturer of all test compounds including DMSO.

Section 2.2 – Were the animals weighed daily to calculate dose? What time were the animals dosed?

Section 2.3 – Thyroid hormone follow a diurnal pattern. What time was the blood collected for these measures?

Lines 146 -147 – I believe this increase in body weight was for all F-53B treatment groups for both sexes? The line graph is a little hard to read, and it is suggested to make the data really clear in the text and then edit the graph if possible (the fonts and shapes are quite small).

Figure 4 – I don’t think it was ever described in the Methods how expression of TSHR and TPO were calculated? Was this based on DAB intensity alone? What program was used? How many sections were analyzed per animal? Analyses such as these can be powerful but also prone to artifacts. It is suggested that the methods be very clearly stated so that another laboratory could repeat the experiment.

Figure 4 – This comment could be an artifact of the proof I received, but the IHC images look to be of low resolution and they’re very difficult to interpret. Please ensure high resolution images are re-submitted; additionally, you could also stack panel B underneath panel A and then resize the images/graphs fit a double-column width. The graphs have a very small font and are also difficult to read, so this re-size could help both problems (images and graphs).

Line 189 – Please cite the specific OECD studies. A suggestion is to also include new DART guidelines that also suggest inclusion of thyroid hormone measures in adult, pregnant, and lactating animals (e.g. OECD 414, 421, 443, etc).

Discussion – The discussion does not do the study justice; there are a lot of interesting data points that could be discussed to further the impact of this work. Hypothyroxinemia is a hallmark of several PFASs, so it is interesting that this study reports similar findings, but the authors only cite two other studies related to this. Also, it does appear that exposure to F-53B may induce sexually dimorphic biological responses, which is also a hallmark of PFASs. I suggest a more thorough synthesis of these findings within the context of PFAS research would greatly improve the paper.

Line 216 – In my opinion these results do not demonstrate that F-53B is accumulating, as no toxicokinetic data are presented. If there are no serum/tissue concentrations to present I believe this statement should be deleted or further justified.

Minor Comments

Lines 41-50 – It may be a little confusing for readers to visualize how 6:2 Cl-PFAES is related to F-53B. It may be helpful to include a figure of their structures, molecular weight, etc. In this figure the Authors could also include other sulfonated PFAS like PFOS and PFHxS for comparison purposes.

Line 112 – Was the solvent for your peroxidase block water, PBS, or methanol?

Line 112 – I recommend just deleting the beginning words “Then, “and write the sentence as “The sections were then washed twice with…..”

Supplementary file – The supplementary file I received only contained information identical to Table 1. This may have been an accidental upload error?

Author Response

Thanks for your comments.

Because of your specialized revision, our manuscript improved sufficiently.

We try to do our's best to reply sincerely and provide a point-by point response.

Please find attached detail for more information.

Once more thnk you for your detailed comments.

Best regards.

Reviewer 2 Report

Comments:

In this study, the authors study the thyroid toxicity of 6:2 Cl-PFAES in rats and find 6:2 Cl-PFAE is a strong regulator of thyroid hormones in rats. The manuscript is well written and organized. It seems to me that this is the first study of 6:2 Cl-PFAES on thyroid dysfunction in rats, but this innovation has not been addressed in the main text. Besides, the discussion section needs to be expanded well in the revised manuscript before the publication.

Specific comments

  1. Line 49: reference 12 seems not right. It is related to the topic of freshwater alga, but the authors are talking about the human serum and fish muscle.

  1. Line 53: this sentence needs a reference.

  1. Line 56: reference 17 is not right here. Something might go wrong with the edits on the references. Please check the citation throughout the paper.

  1. Lines 51-57: the section on the study propose of this study should be improved. Is there any other toxicity that has been studied before except thyroid dysfunction? As I know, 6:2 Cl-PFESA is well known to have hepatotoxicity effects.

  1. Lines 92- 98: it seems several endpoints have been evaluated in this study but most of the results are not included in the main text or the supporting information. Please include these results though they might be insignificant or remove these unnecessary descriptions here.

  1. Lines 100-105: please see comment 5.

  1. Results, line 128-: the sexual difference is not discussed in the result section. For example, as shown in Figure 1C, serum TSH levels in female rats are 2 times lower than those in males. Anyway, if the authors tend to present the data in the groups of males and females (Figures 1, 2, 3 and 4), the sexual difference should be addressed in some ways.

  1. Figure 4: the resolution of this figure is too low for readers to recognize.

  1. Lines 195-204: the comparison between the thyroid toxicity of 6:2 Cl-PFAES and that of PFOS should be added here. This will be helpful for readers to understand the importance of the thyroid toxicity of 6:2 Cl-PFAES.

  1. Lines 205-206: the species should be mentioned here. Zebrafish?

  1. Lines 261-262: the format of this reference is not right. I have not looked into each reference. Please check the format of references and revise accordingly.

Author Response

(The authors gave the same response as above.)

Round 2

Reviewer 1 Report

Please see attached word file; not all of the original concerns were addressed in the revised manuscript. 

Author Response

(The authors gave the same response as above.)
